# Taste Modulator Influences Rare Case of Color-Gustatory Synesthesia

**DOI:** 10.3390/brainsci9080186

**Published:** 2019-07-31

**Authors:** Catherine Craver-Lemley, Adam Reeves

**Affiliations:** 1Department of Psychology, Elizabethtown College, Elizabethtown, PA 17022, USA; 2Department of Psychology, Northeastern University, Boston, MA 02115, USA

**Keywords:** color-gustatory synesthesia, taste, taste modulator, synesthesia

## Abstract

We investigated the effect of a sweetness blocker on the synesthetic taste experience of a rare color-gustatory synesthete, E.C., for whom specific colors elicit unique tastes. Blocking E.C.’s sweetness receptors while the tongue was otherwise unstimulated left other taste components of the synesthesia unaltered but initially reduced her synesthetic sweetness, which suggests a peripheral modulation of the synesthetic illusion.

## 1. Introduction

Synesthesia occurs when the stimulation of one sensory modality (the “inducer”) triggers an involuntary and simultaneous perception in the same or in another modality (the “concurrent”) [1]. Synesthesia can involve cross-linkage among any of the sense modalities. For example, sound-color synesthetes may see colored shapes in response to hearing sounds such as music or voices, and grapheme-color synesthetes see unique colors when viewing written words, letters or numerals. Here we report on “E.C.”, a highly unusual case of color-gustatory synesthesia, in which color acts as the inducing stimulus for concurrent gustatory sensations. In the case of the slightly less unusual lexical-gustatory synesthesia, written or spoken words elicit taste sensations [2,3,4,5,6,7,8]. As color is frequently the concurrent [9], but rarely the inducer [10], our case, in which color is the inducer, is also untypical in this respect. Indeed, we are aware of only two similar cases, “S” and “T.K.”, reported a century apart [11,12]. S experienced both sound-color synesthesia and a reportedly less intense color-gustatory synesthesia [11]. S stated “…when I put my mind intently on colors, I taste them. I can taste blue” (pp. 40–41). Little else was described regarding S’s color-gustatory synesthesia in the original report. Fortunately, Nikolinakos et al. [12], provided detailed documentation of TK’s color-gustatory experience, allowing for some comparison with our case.

Craver-Lemley and Mastrangelo [13], had already demonstrated some degree of cognitive modulation of E.C.’s synesthesia, as indicated by her experience in viewing the face/urn display in Figure 1. Viewing this figure, she reported that the blue “tasted very sweet” and the green tasted “fresh, like rain with no humidity, a hint of cilantro, slightly tangy”. Both tastes were never experienced simultaneously; instead, the taste depended upon the color of the figure that was perceived—faces or urn. Her synesthetic tastes ‘flipped’ along with her visual reversals; she only experienced synesthesia for the color that forms the figure. In this respect, her form of synesthesia is generally consistent with lexical-gustatory synesthesia, which also shows cognitive influences [7,8]. For example, Bankieris and Simner [2], showed that particular phonological features of inducer words predict certain categories of taste in the concurrent, although the latter finding reflected sound symbolic patterns in English (onomatopoeia), whereas the former finding (Figure 1) primarily reflects the disposition of attention.

The purpose of the present study was to determine whether E.C.’s color-gustatory synesthesia could also be modulated at a sensory level. We investigated this by determining whether a sweetness blocker, *Gymnema sylvestre*, when applied to the tongue, could influence E.C.’s synesthetic sweetness. E.C. reported that her synesthesia occurs in her mouth and tongue, which is consistent with accounts from other individuals experiencing synesthetic taste [3,4,11,14] Clearly, this does not imply a sensory origin—the synesthetic taste originates as an association in the brain, even if it is referred back to the mouth. However, some form of sensory modulation might be possible. An apparently adequate test by applying a real taste to the tongue while simultaneously inducing a taste could be problematic because E.C. does not experience synesthetic taste blends, as illustrated by Figure 1, rather she experiences one taste or the other, but not a combination. Therefore, the hunt was on to find a taste modulator that could affect the tongue in the complete absence of gustatory stimuli (food or liquid). The sweetness blocker, *Gymnema sylvestre*, was used because it blocks only sweetness, leaving other tastes unaffected.

## 2. Method

### 2.1. Participant

E.C. is a right-handed 27-year-old female graduate student who experiences color-gustatory synesthesia. Whenever E.C. views a specific color, she automatically experiences distinct taste percepts, sometimes accompanied by texture and emotions (Table 1 and Table 2). Her reports are consistent over years, as is generally typical of synesthetes. Sweet is her most frequent synesthetic taste, sour and bitter occur infrequently, having few inducers, and saltiness does not occur at all. Markedly, T.K. also reported an absence of saltiness with his synesthesia [12]. Additionally, E.C. often describes different colors as eliciting a ‘spicy’ taste. She also experiences tactile sensations (such as a “sand paper” or “grainy”) in her mouth when viewing some colors. Metallic colors in Crayola^®^ crayons will induce a “crackling” sensation that she likens to Pop Rocks^®^ candies and finds to be very enjoyable. She reports that her synesthetic taste lasts as long as she views a color. E.C. reports that she was unaware that she had synesthesia until the topic was discussed in one of her classes. She is not aware of any of her relatives being synesthetes.

This research was conducted in accordance with the American Psychological Association’s standards for the ethical treatment of subjects and with the approval of the Institutional Review Board (IRB) for Human Research of Elizabethtown College, IRB #FA09-05. Before participating, E.C. was informed that she could leave the experiment at any time without penalty. She was informed of the full procedure of the experiment, including being asked to describe her synesthesia, and that she would be given health food store teas or powders. She signed the informed consent form approved by the IRB.

### 2.2. Materials

Crayola^®^ crayons were used to induce E.C.’s synesthesia. The approximate R, G, B values provided by the manufacturer for a color monitor in 2009 were Yellow (252, 232, 131), Periwinkle (197, 208, 230), Shocking Pink (251, 126, 253), Metallic FX Steel Blue (0, 129, 171), Metallic FX Big Dip O’ Ruby (156, 37, 66). The *Gymnema sylvestre* tea and a gunpowder green tea (*Camellia sinensis*, used for a placebo condition) were purchased from Starwest Botanicals, Inc. Both loose teas were brewed according to the manufacturer’s instructions and presented at approximately room temperature. We also used a powdered form of *Gymnema sylvestre* purchased in 400 mg capsules from Swanson Health Products. In this case, E.C. spread the contents of two capsules over the surface of her tongue. *Gymnema sylvestre* is a known suppressor of sweetness, but not of other taste sensations, which acts to inhibit sucrose receptors on the tongue [15]. Given that E.C.’s concurrent can be sweet, *Gymnema sylvestre* provides a unique opportunity to test the role of sensory modulation in this form of synesthesia.

### 2.3. Procedure

Across six sessions, over a 16-month period, E.C. was presented with either the sweetness blocker, *Gymnema sylvestre* (tea that blocks sugars from activating sweet receptors on the tongue) or a placebo, *Camellia sinensis* (gunpowder green tea) that looks and tastes like the experimental tea. The blocker was given in four of the sessions and the control tea was given in the remaining two sessions. Immediately prior to each session, she was also asked to provide a report of the tastes elicited by the crayons after swishing distilled water around her mouth to provide a baseline. During the first, second, and fourth experimental session, E.C. was instructed to swish an ounce of the tea around the inside of her mouth for 30 s and then to swallow. On the third experimental session, *Gymnema sylvestre* powder made from ground tea leaves was placed directly on E.C.’s tongue instead of having her drink the tea. After she ingested the tea, she was individually presented with five different colors (see above) in random order. These colors were selected because pre-testing revealed that they typically elicit the sweetest synesthetic tastes for E.C., and because none of these colors evoke an aversive taste experience for her. Using crayons also allowed us to include stimuli that trigger the “crackling” sensation, along with taste. She reported that she was familiar with the tastes associated with the crayon colors yellow, periwinkle, and shocking pink, but that the crackling sensations and colors associated with the metallic ruby and steel blue crayons were novel.

In all conditions, after each color was presented, E.C. rated the intensity of her synesthetic sweetness on a scale from 0 to 5 (0 = none; 1 = weak to 5 = strong). She also described her synesthetic experience while viewing each color. Upon the conclusion of each session, E.C. was given chocolate candies to consume. After the *Gymnema sylvestre* sessions, she reported that the candy tasted bland, “really strange” and not sweet, verifying that the *Gymnema sylvestre* affected her “normal” taste sensation (as it does with non-synesthetes).

At the start of the fourth experimental session, we asked E.C. to memorize the codes for the five experimental colors until she could evoke a mental image of each color when she heard the associated code. We then asked her to generate a mental image of each color and to rate the sweetness of the concurrent induced by the mental image. After a 30-min break, she then participated in the same baseline versus *Gymnema sylvestre* comparison as was run in the previous three experimental sessions. The sessions took 40 min to an hour to run.

## 3. Results

E.C. initially provided descriptions and ratings for the experimental condition that were less sweet, indicating that her synesthetic sweetness was modified by the *Gymnema sylvestre* (Table 2). Her textural experience was not affected. E.C. appeared disconcerted that her colors were not eliciting her familiar tastes. E.C. described her experience as “like seeing a little girl that you know, opening her mouth to speak and hearing an old man’s voice instead.” The *Gymnema sylvestre* tea diminished or eliminated the synesthetic sweetness E.C. typically experiences when viewing these colors. The colors still triggered some taste and texture, however, indicating that the tea influenced the sweetness only of E.C.’s synesthetic experience.

Although the first two presentations of the *Gymnema sylvestre* tea impacted E.C.’s synesthesia, while the placebo did not, when we later placed *Gymnema sylvestre* powder directly on E.C.’s tongue (this occurred a month after the second presentation of the tea), it had no effect. We thought that perhaps the powder method of administering the *Gymnema sylvestre* might account for the difference, which may be less effective [15], so we tested her three weeks later with the *Gymnema sylvestre* tea, and again there was no effect. Critically, her descriptions of the tastes, not just her sweetness ratings, no longer differed between baseline and tea. Figure 2 shows this result by plotting her sweetness ratings in the baseline and blocker conditions across experimental sessions 1 through 4. A two-way ANOVA of the session by the condition (baseline or blocker) confirmed the visual impression given by Figure 2, in that the main effects of the session (F_3,32_ = 36.1, *p* < 0.05) and condition (F_1,32_ = 31.5, *p* < 0.05) were significant, as was their interaction (F_3,32_ = 10.2, *p* < 0.05). The significant interaction is accounted for by the elimination of the blocker effect in sessions 3 and 4.

We also found that E.C.’s mental images of the experimental colors yielded the corresponding synesthetic tastes. Previously, E.C. had reported that, while imagined colors could trigger her synesthesia, the mental imagery only elicited a “ghost” of her typical synesthetic taste. However, her synesthetic sweetness ratings due to imagery were somewhat higher than the baseline ratings she provided when actually viewing the crayons. A summary of these findings is provided in Figure 3, which plots the mean sweetness ratings averaged over the baseline conditions, in the first two and last two blocker sessions, and with imagery. This finding is in line with other reports provided by synesthetes, where the mental image of an inducer can sometimes trigger a more vivid synesthetic experience than the original inducer [16] (p. 198), but it is in contrast to Nikolinakos et al.’s [12] subject, T.K., whose mental images of colors did not produce his gustatory synesthesia.

## 4. Discussion

We found that the *Gymnema sylvestre* tea suppressed E.C.’s synesthetic sweetness only at the time of the first and second presentations. One way of explaining the reduction in the intensity of the concurrent after repeated administration of the sweetness blocker is in terms of E.C.’s expectations, or desire to please the experimenter [17]. However, E.C. was given the green tea in the very first of her sessions, and the tea had no effect. The blocker was given in subsequent sessions, and, as mentioned, it tasted like the green tea. Therefore, the strong reduction in the synesthetic taste in the first experimental session (see Figure 2) is unlikely to be an expectancy effect.

We therefore consider possible physiological effects. One is that spontaneous neural activity generated by the tongue in the absence of stimulation (a tonic ‘bottom-up’ signal) is needed to activate the gustatory cortex sufficiently for synesthetic taste sensations to occur. That is, sweetness cells in the taste buds must be spontaneously active for the color impression to be able to generate a concurrent taste through a synesthetic process. Spontaneous activity is known to occur; for example, 82% of taste receptors (fungiform papillae) on the tongue of the rat are spontaneously active [18]. The hypothesis that such spontaneous activity supports gustatory synesthesia is novel, and not based on known physiology. If true, a blocker such as *Gymnema sylvestre*, which decreases spontaneous activity in sweetness cells, would attenuate the spontaneous bottom-up signal and thereby eliminate the concurrent taste. An additional possibility is that the blocker may have affected the salivary response of the tongue and palette that is normally associated with sweetness, indirectly modulating the synesthetic sweetness. Other explanations may be possible. However, any explanation must also account for why E.C.’s other synesthetic tastes (bitter and sour) were not affected. The colors associated with those tastes continued to have their normal effects throughout testing.

If any such physiological account is correct, why did repetition in experimental sessions 3 and 4 restore the concurrent? The direct effect of the blocker on the sweetness of the candy bar that E.C. ate at the end of each session did not adapt out—the blocker was effective at the start of the experiment and at the end. It therefore seems unlikely that the effect of the blocker on the synesthetic response adapted out. A hypothesis we favor is that the long-established color taste associations in memory, as revealed in the imagery test in the last session, eventually over-ruled the synesthetic effect of the blocker. If so, E.C.’s synesthesia may be unique in demonstrating a change in the control of synesthesia over repeated administrations from more sensory control to more cognitive control, a change not previously mentioned in the literature possibly because of the difficulty in eliminating the sensory basis of the concurrent, a difficulty removed for E.C. by the fortuitous availability of *Gymnema sylvestre*.

A basic issue in the study of synesthesia concerns the relationship between the concurrent, which is by definition imaginary, and everyday mental images. Craver-Lemley and Reeves [16], argued that they are distinct, as concurrents are obligatory, experienced only in the presence of the inducer, and of uncontrollable vividness, whereas mental imagery is typically voluntary, can be elicited by instruction in the absence of stimulation (except for after-images), and, to the degree indicated by Gordon’s test of controllability of mental imagery, can have the content one wishes and be as vivid as one can achieve. Critically, synesthetes themselves distinguish between everyday mental images, which they can summon at will, and concurrents, which are firmly attached to specific stimuli (as illustrated for E.C. in Table 1). It is therefore of special interest that a mental image of a crayon color could, just like the crayon itself, elicit a taste in E.C. Yet this interaction is perhaps less clear-cut than it seems. Although E.C. did not know the names of the crayons at the start of the experiment, she had to learn them in order to generate appropriate imaginary colors when tested during the final session. To learn them, she had to see the crayon color while hearing its name (“periwinkle”, etc.). Therefore, the route from mental image to concurrent may be via a learnt association with the name, rather than a direct effect. Evidence for a direct (non-verbal) link might be that E.C. experienced a brand-new sensation when presented with a compound of two or more distinct inducers at the same time. However, we have never found a compound that induces a new sensation; instead, we found that E.C. alternates between experiencing the concurrent due to each part of the compound, as illustrated in Figure 1. We therefore believe that the relation between mental imagery and synesthetic imagery needs further study.

An important further issue is whether synesthetic and real sensations have similar consequences. For example, Chiou et al. [19], when studying lexical-color synesthesia, found that real and synesthetic reds and greens, although equally vivid, differentially biased binocular rivalry. Real colors induce a localized sensory-level color-opponent (sensory) bias, whereas synesthetic colors induce a non-localized color-congruent bias, which Chiou et al. interpret as a cognitive effect. In the case of E.C., we can only report that real and synesthetic sweets, sours, and bitters seem similar to her, but unfortunately, we have no evidence to decide if this equivalence in experience translates into sensory substitution. For example, do synesthetic sweets adapt the way that real ones do? And if so, does the blocker affect their adaptations equally?

## 5. Conclusions

Our conclusions are based on data from a single, very rare, subject, and are speculative. This paucity of cases should not detract from the theoretical importance of E.C.’s synesthesia, however, in which—as is otherwise almost never the case—an all-or-nothing sensory manipulation could be applied to the concurrent. We hope that more cases of this sort can be found, so that the necessity or otherwise of an active bottom-up ‘tonic’ signal can be more firmly established.

## Figures and Tables

**Figure 1 brainsci-09-00186-f001:**
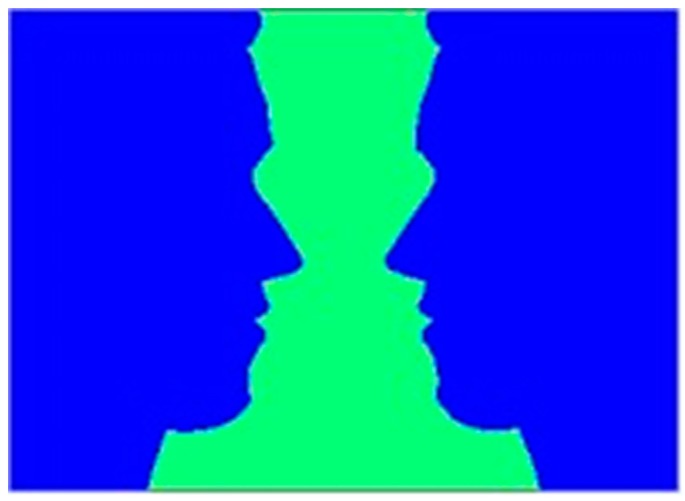
A color-gustatory synesthete, E.C., reported that the blue “tasted very sweet” and the green tasted “fresh, like rain with no humidity, a hint of cilantro, slightly tangy”. In this figure, both tastes were not experienced simultaneously. The taste depended upon the color of the figure that was perceived—faces or urn. Her synesthetic tastes ‘flipped’ along with her visual reversals.

**Figure 2 brainsci-09-00186-f002:**
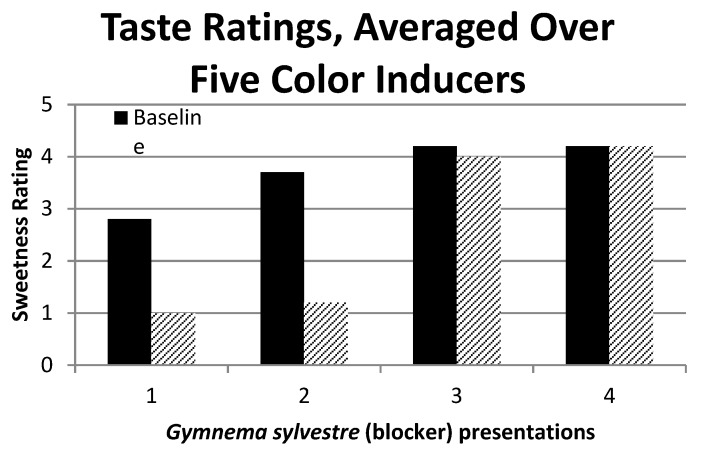
E. C.’s synesthetic sweetness ratings for the baseline and sweetness blocker conditions across four experimental sessions over a 16-month period. The blocker stopped influencing the synesthetic sweetness after the first two presentations.

**Figure 3 brainsci-09-00186-f003:**
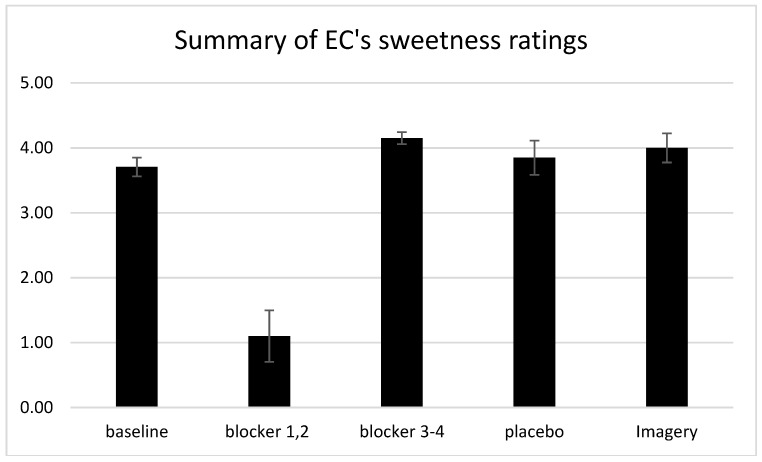
E.C. viewed five color displays and rated the synesthetic sweetness intensity after drinking water (baseline condition; average of six sessions), drinking a taste modulator which blocks sweetness (blocker condition; average of sessions 1–2 and 3–4), or drinking a placebo (average of two sessions), and after simply evoking mental images of the five colors (in experimental session 4). Bars show ±1 standard error of each mean.

**Table 1 brainsci-09-00186-t001:** Illustrative concurrents.

Inducer Color	Synesthete’s Concurrent
Burnt orange	Unpleasant, tastes like bitter salad greens, peppery, too much pepper
Spring green	Fruity and minty
Blue violet	Honey and sugar taste
Brown	Thick cream, heavy whip cream

**Table 2 brainsci-09-00186-t002:** Experimental ratings.

Inducer Color	Baseline Comments	Sweet Rating ^	Post-Sweet Blocker Comments	Sweet Rating ^
Yellow	Very sweet, like cheap grocery store cupcake frosting	3.75	Not as sweet but still unpleasant, flaky, gooey	0
Periwinkle	Like it! Sweet and spicy, maybe tastes like a gardenia would	3.25	Still a favorite	1.75
Shocking Pink	Sweet, but not as overwhelming as yellow, like a Lifesaver^®^ candy—sorbet and mint, no texture, cheerful	3.0	No taste	0.5
Steel Blue	Sweet and spicy, bold, textural experience like Pop Rocks^®^ candy (crackling and popping sensation), pleasant	3.25	Not pleasant, even though the Pop Rocks^®^ texture is there, taste is turned down	2.0
Ruby	More spicy than sweet, tangy, Pop Rocks^®^ experience, energizing	3.0	Decrease in the flavor, but texture is there	1.25

^ Mean ratings from the first two *Gymnema sylvestre* sessions.

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
