# Peer review of "Taste Modulator Influences Rare Case of Color-Gustatory Synesthesia"

_brainsci, 2019, doi:10.3390/brainsci9080186_

Reviewer 1 Report

Review of 2nd draft

An interesting article albeit based on a single case. However, the authors have done well to point this out and have now included a wider discussion of additional related studies. This said, some clarification and editing is needed to make the materials added to the Discussion section of the revised version clearer. Specifically,

1) The inclusion of EC’s bitter and sour tastes in the Discussion section should be omitted as the authors note early that “sour and bitter occur rarely” – so why distract from the main issue- sensation/perception of sweetness? Suggest omitting these comments (e.g., line 227)

2) Paragraph from line 230-244 needs some editing and perhaps rewording to be clearer. Expressions like “blocked the blocker” (line 238) can be improved on for better clarity. Edit line 234 – currently makes little sense as written.

3) Concluding sentence of this same paragraph (see lines 239-244) leaves one still trying to ascertain the author’s overall point here. Perhaps add a sentence (or two) to stress its relevance and connection to overall argument and speculations.

4) More care in using terms like “memory”, “learn” and “cognition” would be helpful in places- e.g., seems at times that some of these terms are used synonymously at times – and certainly they are relevant to the overall discussion.  

5) The use of the word “compound” (lines 264-266) was not clear in reference to “distinct inducers” and perhaps in general sense. Perhaps this can this be edited to improve the descriptions here.

6) Might consider putting paragraph in lines 36-40 (and its related references) back in the article as still seems relevant to the discussion presented, although in a different location (perhaps in the Discussion section).

Overall, suggest modifying this article with just a bit more editing and clarifying in places. An interesting piece of research!

Reviewer 2 Report

The paper has been greatly improved by the review process, although obviously the odd results and more speculative conclusions remain (and would be for others to judge once in the public domain)

Author Response

no second re-review received.

This manuscript is a resubmission of an earlier submission. The following is a list of the peer review reports and author responses from that submission

Round  1

Reviewer 1 Report

This article presents an interesting rare case of color-gustatory synesthesia. One unresolved issue is why the sweetness block stopped being effective after the first two presentations- as the authors note. The explanation provided hinted at the effect of mental images and memory. It would be interesting to know if there any other studies which show this type of change in synesthesia reactions in other forms of synesthesia, where mental images perhaps helped induce a shift from lower to higher synesthesia. I suggest the authors might add a sentence or two in the Discussion section that explicitly states that mental imagery (associated with memory- a top-down phenomenon) could have produced this change and should be looked as a possible explanation for stopping the effectiveness of the sweetness blocker in this case. Overall, although based on a single case study, the article is well constructed.

Author Response

Reviewer 1 thought that the study was ‘well conducted’ and we appreciate this. He or she also thought that the mental imagery task might have produced the change (abolition of the blocker effect). However, imagery occurred only in session 4, so it doesn’t account for the change already occurring in session 3.  We have added in this point, which is a good one.

Reviewer 2 Report

The paper describes a case study of a female colour-taste synaesthete. Over several sessions, the participant either ingested a tea containing a sweetness taste receptor blocker or green tea (or had taste blocker powder placed on her tongue) and gave subjective ratings of sweetness for concurrents elicited by coloured Crayola pencils. The authors found that in half of the 'sweetness blocker' sessions, the participant reported lower ratings (although no statistical comparison was provided) whereas in the other half and the control sessions, they did not. The authors claim that a change in the sweetness of concurrents after ingestion of a taste receptor blocker would suggest that colour gustatory synaesthesia is of the lower type of synaesthesia.

I have major and minor concerns about the methodology and interpretation in this paper. I organised them as such, below. My main concern is that the paper makes an unjustified assumption about the necessity of activity of taste receptors on the tongue in experiencing phantom tastes (which forms the backbone of the entire paper). The introduction, the aims of the study and the discussion are based on this empirically unsupported assumption. The second major concern is the lack of control conditions in the design of the study and the small number of ratings collected.

Major concerns:

1.   Line 35: "Two main types of synesthesia have been identified …." the authors should rephrase this in a way that makes it clear that "types" here means synaesthesia type in terms of its underlying neural mechanism) and not types as in variant of synaesthesia. The author may also want to give examples of both these types, so that the reader can evaluate whether the author's argument in the paragraph starting on line 47 is sound.

2.   Line 35: This whole paragraph should be unpacked - not enough information is given.

·      "Recent studies with lexical" - studies carried out in 2009 and 2006 are by no means recent. Author should change to "existing studies".

·      "synesthesia can influenced by a symbolic/conceptual level of representation" - The sentence here is quite vague and the reader expects an example or explanation. In what way? Can the author provide examples?

3.   Line 47: "If her synesthetic taste is affected by the blocker, then her type of synesthesia can be modulated at a sensory level, indicating that it is of the “lower” type. " This argument is unfortunately invalid - to my knowledge, it has never been argued that, for a synaesthesia to be of the lower type, sensory receptors have to be involved in the synaesthetic sensation. It is the neural structures that process these signals - not excitation of the sensory receptors - that are involved in synaesthesia and that elicit the synaesthetic sensation. The concurrent experience is triggered in the brain - as opposed to on the tongue - by signals that travel along connections from other parts of the brain. In lower synaesthesia these come from sensory cortices, whereas in higher synaesthesia they come from areas that process conceptual information. The signal does not need to go via the receptors for the concurrent to be experienced, in the same way that in phantom limb pain, the receptors don’t need to exist on the limb for the pain to be felt. Moreover, if experiencing a concurrent required activity in its sensory receptors then synaesthesia could not exist in the blind, but it can. Therefore, showing that tasting or not tasting a concurrent when the receptors are blocked says nothing about whether synaesthesia can be categorized as lower or higher. The authors cannot make the claim that their results suggest anything of the sort.

4.   Did the authors test this participant for consistency of concurrents? If so please include details of this. I assume the authors have this data as they asked the participant to report the tastes of colours before ingestion of the teas/power.

5.   Was the participant only asked to rate sweetness for colours once within each session? Generally, repetitions of trials are needed, particularly in single case studies in order to increase the reliability of the data collected.

6.   There is still a possibility that the participant rated the sweetness of colours based on what she presumed she was ingesting (or having put on her tongue) and this has not been considered or at least discussed. How does the author know that the participant did not simply recognise the control tea as green tea and the Gymnema sylvestre as 'not green tea' when reporting her tastes and chose to report lower sweetness in the tastes when she believed she was being given a taste blocker? It is possible that when the powder was placed on her tongue, she presumed it was not a taste blocker and continued to report tasting sweetness in colours when the power was administered. Is there any evidence that people cannot taste the difference between the two teas? What was the participant told they were being given on the different sessions? Please include these details.

7.   It is also impossible to tell (as a result of the study design) whether the participant's ratings of sweetness would have changed for colours that were not sweet because only sweet colours were used as inducers. I am aware that the authors mention that most of the synaesthetic tastes the participant experiences are sweet - was the green in the figure sweet also? If not, this colour could have been used as a control. The authors also mention that EC experienced sour and bitter sometimes, so the colours that elicit these could have been used also. Moreover, the authors could also have taken ratings for other tastes in order to show that the participant did not simply change her ratings depending on session.

8.   Line 123: "We thought that perhaps this method of administering the Gymnema sylvestre might account for the difference." - please explain how this might account for the difference - the paper mentions that the taste blocker took effect on every session (as evidenced by the comments about the chocolate tasting bland).

9.   I cannot comment much on the discussion because it is based on the unjustified assumption that tasting phantom tastes despite having blocked taste receptors suggests that colour synaesthesia is of the lower type, and that the sensation experienced as a result of an inducer requires taste receptors at all. This means that the claims that colour taste synaesthesia moves between being higher and lower is unsupported by the data and findings.

10.     One comment I do have is that the results are inconclusive given that the participant gave lower ratings on only 50% of the sessions. (and no statistics have been provided to support that the difference in ratings was significantly different than 0)

Minor issues:

1.         Line 24 - "In the slightly more common case of lexical gust" this should be rephrased. It is unclear that the author is comparing lexical gustatory to colour gustatory - it seems that they are comparing to the different synaesthesias mentioned in the previous paragraph and that they are saying that lex gust synaesthesia is more common.

2.         Line 26-28: "Since colour is frequently the concurrent …." cumbersome sentence, perhaps rephrase.

3.         Line 27 - "atypical" not "untypical"

4.         Line 28 "Indeed, we aware" - missing "are"

5.         Line 2 - "We investigated this by determining how a taste modulator," - please explain what this taste modulator does (i.e. does it modulate all tastes? How does it do it? Etc. - or at least copy information from line 84) -

6.         Please include mean ratings in the results section. and error bars on the graph which uses averages of values over sessions.

Author Response

Reviewer 2 wrote a detailed and highly useful review. We hope to have satisfied him or her, but are quite willing to make further changes should they be needed. Meanwhile, thanks for the effort!

I have major and minor concerns about the methodology and interpretation in this paper. I organised them as such, below. My main concern is that the paper makes an unjustified assumption about the necessity of activity of taste receptors on the tongue in experiencing phantom tastes (which forms the backbone of the entire paper). The introduction, the aims of the study and the discussion are based on this empirically unsupported assumption. The second major concern is the lack of control conditions in the design of the study and the small number of ratings collected.

 We do not think the assumption is unnecessary; and the reviewer did not provide an alternative, other than critiquing the experiment, which is fair game (but we think, doesn’t explain the main result.) So we are sticking with it, odd as it seems. We do provide a new reference, for spontaneous activity in taste receptors, which is one essential part of the hypothesis.

Major concerns:

1.     Line 35: "Two main types of synesthesia have been identified …." the authors should rephrase this in a way that makes it clear that "types" here means synaesthesia type in terms of its underlying neural mechanism) and not types as in variant of synaesthesia. The author may also want to give examples of both these types, so that the reader can evaluate whether the author's argument in the paragraph starting on line 47 is sound.

  We re-phrased our argument, dropping the terms upper and lower, as explained in 3. below. We don’t have independent evidence about the neural pathways involved; we know only that blocking the sweetness receptors can modulate her gustatory synaesthesia.

2.      "Recent studies with lexical" - studies carried out in 2009 and 2006 are by no means recent. Author should change to "existing studies".  Thanks. Done.

·   Line 47: "If her synesthetic taste is affected by the blocker, then her type of synesthesia can be modulated at a sensory level, indicating that it is of the “lower” type." This argument is unfortunately invalid - to my knowledge, it has never been argued that, for a synaesthesia to be of the lower type, sensory receptors have to be involved in the synaesthetic sensation.

 OK –We thank the reviewer for pointing this out, and hope now to have made our argument (strange as it is) transparent by removing the ‘lower’ versus ‘higher’ distinction.

 It is the neural structures that process these signals - not excitation of the sensory receptors - that are involved in synaesthesia and that elicit the synaesthetic sensation. The concurrent experience is triggered in the brain - as opposed to on the tongue - by signals that travel along connections from other parts of the brain. In lower synaesthesia these come from sensory cortices, whereas in higher synaesthesia they come from areas that process conceptual information. The signal does not need to go via the receptors for the concurrent to be experienced, in the same way that in phantom limb pain, the receptors don’t need to exist on the limb for the pain to be felt. Moreover, if experiencing a concurrent required activity in its sensory receptors then synaesthesia could not exist in the blind, but it can. Therefore, showing that tasting or not tasting a concurrent when the receptors are blocked says nothing about whether synaesthesia can be categorized as lower or higher. The authors cannot make the claim that their results suggest anything of the sort. Good point. We hope that our argument is now clearer. Synesthesia can be modulated by cognitive or by sensory signals; we give evidence for both. The sensory signal that we invoke is the random neural activity in the sweetness receptors in the (physically unstimulated) tongue, activity that is damped down by the blocker. We have removed the terms upper and lower.

4.   Was the participant only asked to rate sweetness for colours once within each session? Generally, repetitions of trials are needed, particularly in single case studies in order to increase the reliability of the data collected.  Ratings were collected 5 times, once for each of the 5 colors, in each condition. This is stated in the Procedure section. So there were n=10 observations (ratings) collected in experimental sessions 1 and 2, for example.

6.   It is also impossible to tell (as a result of the study design) whether the participant's ratings of sweetness would have changed for colours that were not sweet because only sweet colours were used as inducers. I am aware that the authors mention that most of the synaesthetic tastes the participant experiences are sweet - was the green in the figure sweet also? If not, this colour could have been used as a control. The authors also mention that EC experienced sour and bitter sometimes, so the colours that elicit these could have been used also. Moreover, the authors could also have taken ratings for other tastes in order to show that the participant did not simply change her ratings depending on session.  This is an important point. As stated, her other colors continued to produce their regular synesthetic tastes (sour and  bitter) throughout the experimental series.

8.   I cannot comment much on the discussion because it is based on the unjustified assumption that tasting phantom tastes despite having blocked taste receptors suggests that colour synaesthesia is of the lower type, and that the sensation experienced as a result of an inducer requires taste receptors at all. This means that the claims that colour taste synaesthesia moves between being higher and lower is unsupported by the data and findings. We offered this idea as a hypothesis to explain the results, as we have no other good explanation, but not as a proof, and we hope that this is now clear.

10.     One comment I do have is that the results are inconclusive given that the participant gave lower ratings on only 50% of the sessions. (and no statistics have been provided to support that the difference in ratings was significantly different than 0) Statistics (ANOVAs and error bars) now provided.

Minor issues: all these have been attended to. Thanks so much for pointing them out

Reviewer 3 Report

This is an interesting case study of a rare type of synaesthesia.  The rationale for the study is sound, although the results are not overly compelling because the findings ‘flip’ after multiple testing sessions (from being affected by a taste blocker to not being affected).  The authors offer an explanation for this in terms of a change from bottom-up to top-down influences but it is necessarily rather post-hoc.  As such I would classify the results as anecdotal but nevertheless of interest as nobody really attempted to explore this before.

The higher/lower distinction doesn’t really have much common currency, at least as a binary division.  I think you just need to discuss that there is heterogeneity in what can act as inducer  and, even for the same stimulus, it could conceivably range from physical properties through to linguistic characteristics. 

The introduction needs to describe what is known about this taste modulator which will not be known about to most readers.  For instance how low is “low”?  I suspect that it feeds forward to cortical responses to tastes and it is known that mental imagery of flavors can activate such regions (Simmons, W.K., Martin, A., & Barsalou, L.W. (2005). Pictures of appetizing
foods activate gustatory cortices for taste and reward. Cerebral Cortex, 15,
1602 – 1608)

Author Response

Reviewer 3.

(This is) an interesting case study of a rare type of synaesthesia.  The rationale for the study is sound, although the results are not overly compelling because the findings ‘flip’ after multiple testing sessions (from being affected by a taste blocker to not being affected).  The authors offer an explanation for this in terms of a change from bottom-up to top-down influences but it is necessarily rather post-hoc. 

Yes. We now state that the conclusions are speculative.

As such I would classify the results as anecdotal but nevertheless of interest as nobody really attempted to explore this before.

The higher/lower distinction doesn’t really have much common currency, at least as a binary division.  I think you just need to discuss that there is heterogeneity in what can act as inducer, and, even for the same stimulus, it could conceivably range from physical properties through to linguistic characteristics. 

Yes, we removed higher/lower, and have adopted the terms ‘cognitive’ and ‘sensory’ as these do seem applicable to EC’s case. We hope that this is clear. (We haven’t discussed the idea of a ‘range’ form physical to linguistic, though, as – to us --  this distinction remains categorical.)

The introduction needs to describe what is known about this taste modulator which will not be known about to most readers.  For instance how low is “low”?  I suspect that it feeds forward to cortical responses to tastes and it is known that mental imagery of flavors can activate such regions (Simmons, W.K., Martin, A., & Barsalou, L.W. (2005). Pictures of appetizing
foods activate gustatory cortices for taste and reward. Cerebral Cortex, 15, 1602 – 1608)

An interesting reference ! However we have not discussed it, as we have no reason to think this refers  to the same pathway as in EC’s case (though we don’t know).